# A high-throughput screen identifies inhibitors of the interaction between the oncogenic transcription factor ERG and the cofactor EWS

Taylor R. Nicholas[1], Jingwei Meng[2], Benjamin M. Greulich[3], Teresa Stevie Morris[3], Peter C. Hollenhorst[3]*

1 Department of Biology, Indiana University, Bloomington, Indiana, United States of America, 2 Chemical Genomics Core Facility, Indiana University School of Medicine, Indianapolis, Indiana, United States of America, 3 Medical Sciences, Indiana University School of Medicine, Bloomington, Indiana, United States of America

* pchollen@iu.edu

**Data Availability Statement:** All relevant data are within the manuscript and its Supporting Information files.

## Abstract

Aberrant expression of the transcription factor ERG is a key driving event in approximately one-half of all of prostate cancers. Lacking an enzymatic pocket and mainly disordered, the structure of ERG is difficult to exploit for therapeutic design. We recently identified EWS as a specific interacting partner of ERG that is required for oncogenic function. In this study, we aimed to target this specific protein-protein interaction with small molecules. A high-throughput screening (HTS) strategy was implemented to identify potential protein-protein interaction inhibitors. Secondary assays verified the function of several hit compounds, and one lead compound inhibited ERG-mediated phenotypes in prostate cells. This is the first study aimed at targeting the ERG-EWS protein-protein interaction for the development of a small molecule-based prostate cancer therapy.

## Introduction

Transcription factors are key modulators of cancer. In many cancers, transcription factor function is altered through mutation. Additionally, changes in signaling converge on transcription factors which causes cancer-supporting alterations in the transcriptome. Therefore, understanding how to target transcription factors is essential for advancing the repertoire of cancer therapies. Traditionally, transcription factors are difficult to specifically inhibit as they lack hydrophobic enzymatic pockets and often contain intrinsically disordered regions (IDRs) [1, 2]; however a new area of research aims to exploit other features for inhibition, like specific protein-protein interactions essential for transcription factor function [3].

The majority of prostate cancer is driven by gene rearrangements that result in the expression of certain ETS transcription factors that are normally silent in prostate cells [4]. The ETS transcription factor family is composed of 28 members, some of which are normally expressed in the prostate and control healthy function [5]. However, rearrangement-induced expression of the specific ETS factors ERG, ETV1, ETV4, and ETV5 drive an oncogenic gene expression

**Funding:** Research reported in this publication was supported by the National Cancer Institute (www.cancer.gov) of the National Institutes of Health under Award Number R01CA204121 (P.C.H.), and with support from the Indiana Clinical and Translational Sciences Institute (indianactsi.org) funded, in part by award number UL1TR002529 from the National Institutes of Health, National Center for Advancing Translational Sciences, Clinical and Translational Sciences Award (T.R.N.). Research reported here was also funded by the IU Simon Cancer Center (www.cancer.iu.edu). The funders had no role in study design, data collection and analysis, decision to publish, or preparation of the manuscript.

**Competing interests:** The authors have declared that no competing interests exist.

program in this ectopic setting [4, 6–8]. ETS factors are classified by the presence of a conserved ETS DNA binding domain, so targeting the ETS factor-DNA interaction will likely impact endogenous ETS factors, some of which act as tumor suppressors [6]. Outside of the structured ETS domain, ETS factors are largely disordered, posing a challenge to rational structure-based therapeutic design. Therefore, understanding the specific mechanisms that oncogenic ETS use to drive cancer is essential for the development of precision medicines.

We recently reported that oncogenic ETS factors such as ERG require an interaction with the RNA binding protein EWS to attain oncogenic function [9]. Expression of a point mutant of ERG that disrupts the interaction with EWS results in a significant decrease in prostate cancer cell migration, clonogenic growth, and anchorage independent growth as well as decreased tumor formation in mice [9]. This interaction is specific to oncogenic ETS factors—ETS factors endogenously expressed in the prostate do not interact with EWS. Therefore, the oncogenic ETS-EWS interaction is a suitable candidate for the development of a specific inhibitor for clinical use.

In this study, we aimed to develop a strategy to chemically inhibit the oncogenic ETS-EWS interaction. We specifically chose to target the interaction between ERG and EWS, since ERG is the most commonly rearranged ETS factor in prostate cancer. Using HTS we identified small molecule inhibitors of the ETS-EWS interaction. We then performed secondary interaction and functional assays for hit-to-lead generation. This is the first study aimed at developing inhibitors of this particular protein-protein interaction. Our results suggest that the ERG-EWS transcription factor-coactivator interaction is able to be inhibited by small molecules and the AlphaScreen (PerkinElmer, Waltham, MA) system is a feasible approach for identifying these inhibitors.

## Materials and methods

### Construct cloning and protein purification

ERG was purified as previously described [10]. Briefly, ERG was cloned into pET28a (Novagen) and expressed in BL21 E. coli. ERG was purified with Ni-NTA resin (Qiagen) and eluted using Imidazole. FLAG-EWS 1-355aa was cloned into pGEX-6p-2-GST-EWS after removal of the full length EWS gene. EWS 1-355aa was purified as previously described [9]. Briefly, EWS 1-355aa was expressed in BL21 E. coli, purified using glutathione agarose beads (Pierce), and eluted using 50 mM reduced glutathione. Protein concentration was measured by Bradford assay and by comparing band intensity to a standard curve on a coomassie stained SDS-PAGE gel. Protein was diluted to appropriate working concentrations in assay buffer (1x PBS + 0.5% BSA) before use.

### Z' determination

Assay quality control was performed using 30 nM of both purified proteins, either purified proteins, or no protein (assay buffer alone), pipetted using a multichannel pipette, into 70 wells of a 384 well plate. Assay plates were then incubated for 1 hr at room temperature. Next, 0.4 μg donor and acceptor resins were added to each appropriate well using a multichannel pipette. Plates were incubated for 1 hr at room temperature in the dark and read using the Envision 2102 Multilabel Plate Reader (Perkin Elmer) using the AlphaScreen protocol with excitation set at 680 nM and emission set at 570 nM. Scatterplots were graphed and Z' robust was calculated as previously [11] after the removal of outliers.

### AlphaScreen

30 nM of each protein in 10 μl volume was added to plates using MultiFlo FX microplate dispensor (BioTek). Compounds were then stamped from 384 well mother plates to assay plates

using a Freedom Evo Liquid Handling System (Tecan). Assay plates were then incubated for 1 hr at room temperature. Next, 0.4 μg donor and acceptor resins were added to each appropriate well in 10 μl volume using the MultiFlo FX microplate dispenser and assay plates were incubated for another 1 hr at room temperature in the dark. Plates were read using the Envision 2102 Multilabel Plate Reader (PerkinElmer) using the standard AlphaScreen protocol (PerkinElmer) with excitation set at 680 nM and emission set at 570 nM.

## Affinity pull down assay

Roughly 5 μg of purified His-tagged ERG was diluted in 300 μl binding buffer (100mM sodium phosphate pH 8.0, 600mM NaCl and 0.02% Tween) and incubated with 2.5 μl His-tag isolation dynabeads (LifeTechnologies) for 1 hr at 4 degrees. ERG conjugated beads were washed twice with 700 μl binding buffer to remove unbound protein and then blocked with BSA in NP-40 lysis buffer (50mM Tris-HCl pH 7.4, 250mM NaCl, 5mM EDTA, 10mM NaF, and 1% Nonidet P-40). 14 μg of PC3 nuclear extract (Santa Cruz Biotechnology) was added and incubated for 2 hrs at 4 degrees. Samples were washed twice with 700 μl NP-40 lysis buffer to remove unbound protein. Compounds were then added at designated concentrations and incubated for 1 hr at 4 degrees. After washing four times with NP-40 lysis buffer, proteins were eluted in SDS loading dye, separated using SDS-PAGE gels, and transferred to nitrocellulose membranes. ERG was visualized using Ponceau staining (0.1% in 5% acetic acid) and an interaction with EWS was measured by immunoblot. Purified EWS input was measured by standard coomassie quick staining of SDS-PAGE gels.

## Cell culture and antibodies

RWPE1 and VCaP cells were obtained from ATCC and grown according to ATCC guidelines as follows: RWPE1 and RWPE1-ERG cells were grown in Keratinocyte SFM (ThermoFisher) and VCaP cells in Dulbecco's modification Eagle media (Sigma) with 10% fetal bovine serum (Sigma). RWPE1-ERG cells were created using retroviral transduction with ERG expressed under control of the *HNRNPA2B1* promoter as previously reported [9]. Both RWPE1 and VCaP lines were authenticated by the University of Arizona Genetics Core using Powerplex 16HS Assay (Promega). All cell lines used have tested negative for mycoplasma using the Mycoplasma Detection Kit (Sigma). The EWS antibody (sc-28327, lot number: B0315) is a mouse monoclonal from Santa Cruz Biotechnology. Immunoblot using the EWS antibody was performed at 1:1000. The FLAG antibody a mouse monoclonal from Sigma (F1804, lot number: SLBQ6349V). Immunoblot using the FLAG antibody was performed at 1:1000.

## Migration assay, clonogenic growth assay, and cell viability assay

Migration assays using RWPE1-ERG cells were performed as previously described [8]. Briefly, 50,000 cells were seeded into the transwell insert (8-micron pore size, BD bioscience). Either DMSO or each compound at the indicated concentration was added to the insert and cells were incubated at 37 C with 5% $CO_2$ for 72hrs. Inserts were then removed, stained, mounted, imaged and quantified. Migrated cells are reported as the mean of three biological replicates, each with two technical replicates. Clonogenic growth assays were performed as previously described [9]. 1,000 RWPE1-ERG cells were seeded in each well of a 6 well plate. Cells were incubated for 3 days at 37 degrees Celsius with 5% $CO_2$ prior to the addition of DMSO or 45 μM of each compound. Cells were then incubated for an additional 7 days at 37 C with 5% $CO_2$ before they were fixed with 10% formalin and stained with 0.5% crystal violet in 25% methanol. Stained colonies were imaged and counted using Genesys image acquisition and analysis software (Syngene). Number of colonies are reported as the mean of three biological

replicates, each with two technical replicates. To measure cell viability, 1,500 RWPE1-ERG cells were seeded per well in a 96 well plate. After incubating the cells for 1 day at 37 C with 5% $CO_2$, DMSO or 45 μM of compound was added. Cells were then incubated for an additional 4 days after which MTT reagent (5mg/ml in PBS) was added. After incubation for 4hrs, media was removed and DMSO was added. Absorbance was measured using the ELx8200 plate reader (BioTek Instruments). Cell viability is reported as the mean of three biological replicates, each with four technical replicates.

## Luciferase assays

Luciferase assays were performed as previously described [9]. We used the firefly luciferase reporter pGL4.25 (Promega) driven by the ETS-motif containing *FLH3* enhancer as expression of this reporter has been previously shown to require ERG-EWS [9]. Dual luciferase reportor assay kit (Promega) was used to measure luciferase activity. Relative luciferase activity is reported as the mean of three biological replicates, after normalizing firefly values to renilla values. All biological replicates contain two technical replicates.

## RNA extraction and quantitative reverse transcription PCR

RNA was extracted using the RNAeasy kit (Qiagen). 1 μg of RNA was reverse transcribed using the following 3' primers: 5'-CATGTTGGGTTTGCTCTTCC-3' for ERG, 5'- TCAGA ACCATAGAAGACACC-3' for HSPA8, and 5'-GACTTTGGTTTCCCGGAAGC-3' for 18S. RNA was measured using standard curves as previously described [12] using the 3' reverse transcription primers and the following 5' primers: 5'-ACCATCTCCTTCCACAGTGC-3' for ERG, 5'-CCAACACAGTTTTTGATGCC-3' for HSPA8, and 5'-GGTGAAATTCTTGGACC GGC-3' for 18S. Expression of ERG is normalized to 18S and reported as three biological replicates each represented by the average of two technical replicates.

## Statistical analysis

We performed unpaired t-tests to compare the difference between the DMSO control group and individual Hit treated groups in phenotypic and qRTPCR assays. P values reported above treated groups indicate statistical analyses compared to the DMSO control group. $^* < 0.05$, $^{**}<0.01$, $^{***}<0.001$.

## Results

### HTS set up

We chose to implement the AlphaScreen technology to find small molecule inhibitors of the ERG-EWS protein-protein interaction. AlphaScreens have been used to identify protein-protein interaction inhibitors and are a robust and sensitive method [13, 14]. A pipeline for screen preparation, performance, validation and hit-to-lead generation is detailed in Fig 1.

### EWS 1-355aa interacts with ERG

The sensitivity of HTS requires clean protein preparations to limit assay interference. EWS is difficult to purify in full length because it is prone to aggregation and is rapidly degraded [15, 16]. We therefore sought to identify a more stable fragment of EWS capable of interacting with ERG that could be purified without the presence of degradation products. N-terminal Flag-GST-tagged versions of full-length EWS, or C-terminal or N-terminal regions of EWS were purified and assayed for interaction with His-tagged ERG. The N-terminal fragment, EWS 1-355aa, was able to interact with ERG, better than full length EWS 1-656aa, while the C-

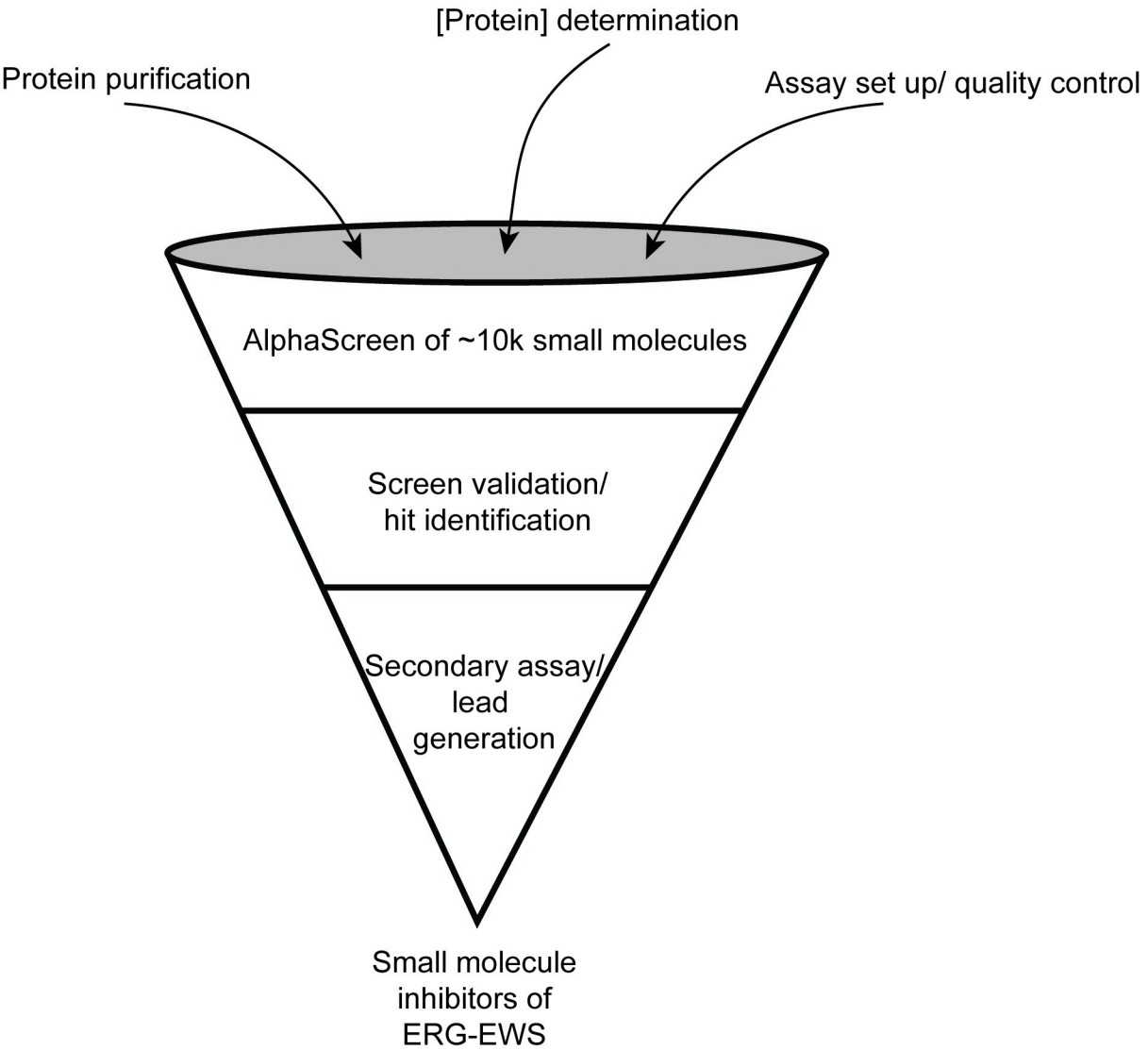

**Fig 1. Pipeline for identifying small molecule inhibitors of ERG-EWS using HTS.**

terminal fragment, EWS 459-656aa was unable to interact (Fig 2A). Interestingly, an N-terminal degradation product arising from full-length EWS, and about the same size as EWS 1–355, also interacted with ERG better than full-length EWS. Based on these findings, we decided to use EWS 1-355aa for HTS.

To determine the concentration of protein to use in the screen, a cross titration was performed using the AlphaScreen beads. Because the proteins were tagged with His and GST, they were directly conjugated to the nickel chelate acceptor and glutathione donor beads, respectively. The concentration of each purified protein was varied while the beads were kept at a constant final concentration of 0.4 μg per well. A matrix of 100, 30, 10, 3, or 0 nM of ERG with 100, 30, 10, 3, or 0 nM of EWS 1-355aa was used to determine optimal signal intensity output. Use of 30 nM of each protein resulted in strong assay signal and was selected for the screen. Lesser concentrations of either protein caused a robust decrease in signal intensity (Fig 2B).

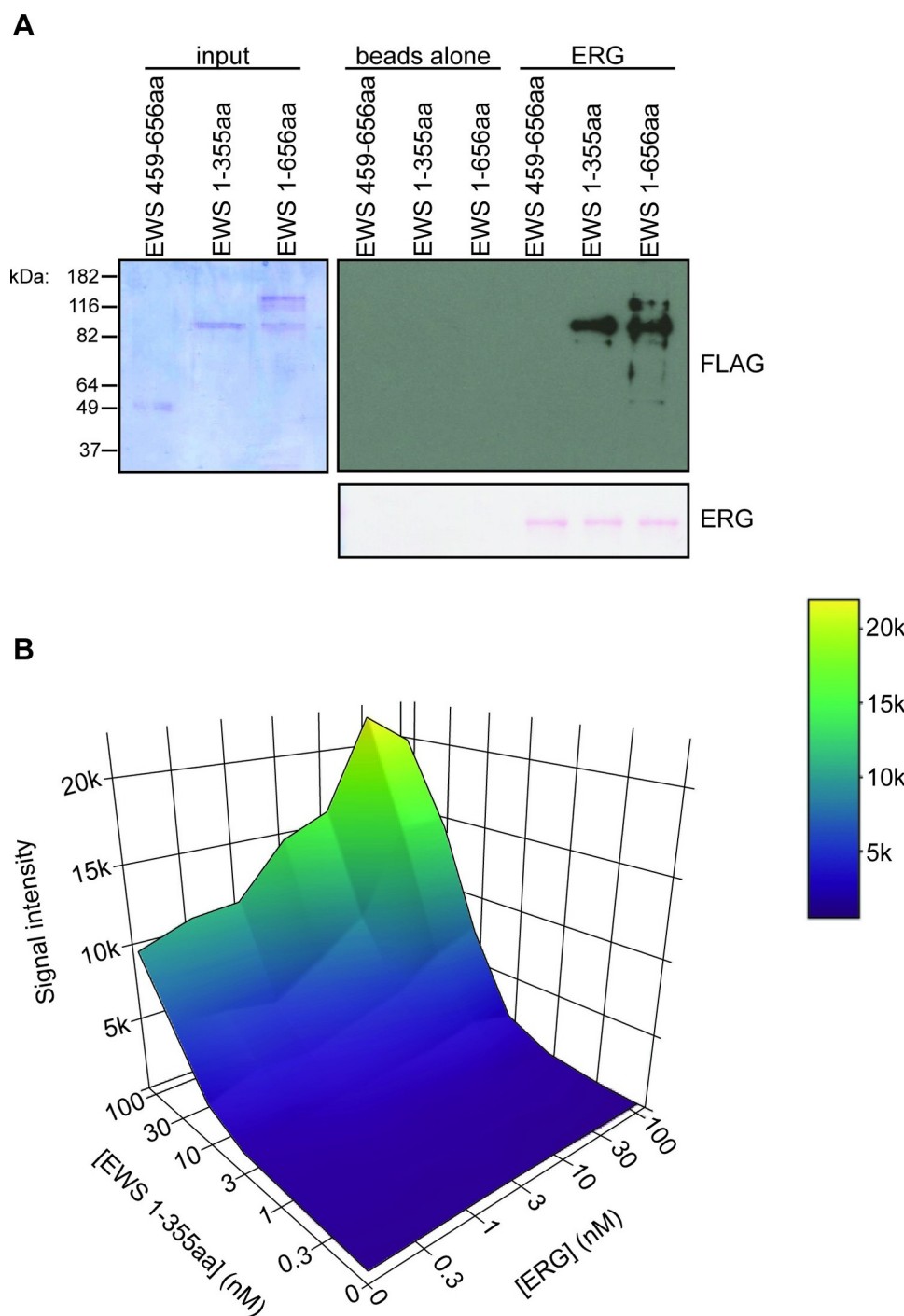

**Fig 2. EWS 1-355aa interacts with ERG.** A) His-tag affinity pull down of purified EWS fragments. Purified His-ERG was conjugated to cobalt beads and used to pull down indicated FLAG-GST-EWS proteins. An interaction with EWS is indicated by the FLAG immunoblot. EWS input is shown as a coomassie stained SDS-PAGE gel. B) Surface plot of cross-titration of ERG and EWS 1-355aa. Concentration of proteins are indicated. Signal intensity is an arbitrary measurement of fluorescence.

## AlphaScreen quality control

Before performing HTS, we determined the AlphaScreen signal separation between positive and negative controls. This is important in determining the ability of the assay reagents or the purified protein to interfere with the instrumentation. 70 replicates of the AlphaScreen were performed using the both proteins conjugated to their corresponding beads (positive control) and the following negative controls: unconjugated beads, ERG conjugated to acceptor beads in the presence of unconjugated donor beads, and EWS 1-355aa conjugated to donor beads in the presence of unconjugated acceptor beads (Fig 3A–3C). The z' robust, an outlier-independent measure of interference that takes into consideration the means and standard deviations of both the positive and negative controls [11, 17], was calculated. A Z' value greater than 0.5 indicates an excellent assay set up [11]. The Z' robust value for no protein was 0.825, for ERG only, 0.846, and for EWS 1-355aa only, 0.725 (Fig 3D), indicating the assay setup allowed minimal interference with the instrumentation.

## AlphaScreen hit generation and validation

We used three small molecule libraries—Lopac 1280, Microsource Spectrum 2400, and Analyticon Natx 5000—for a total of 8680 compounds. Each 384 well assay plate was stamped with compounds from the mother library plates in columns 3–22 to give each compound a concentration of 62.5 μM. Columns 1 and 24 contained assay buffer only. Column 2 contained

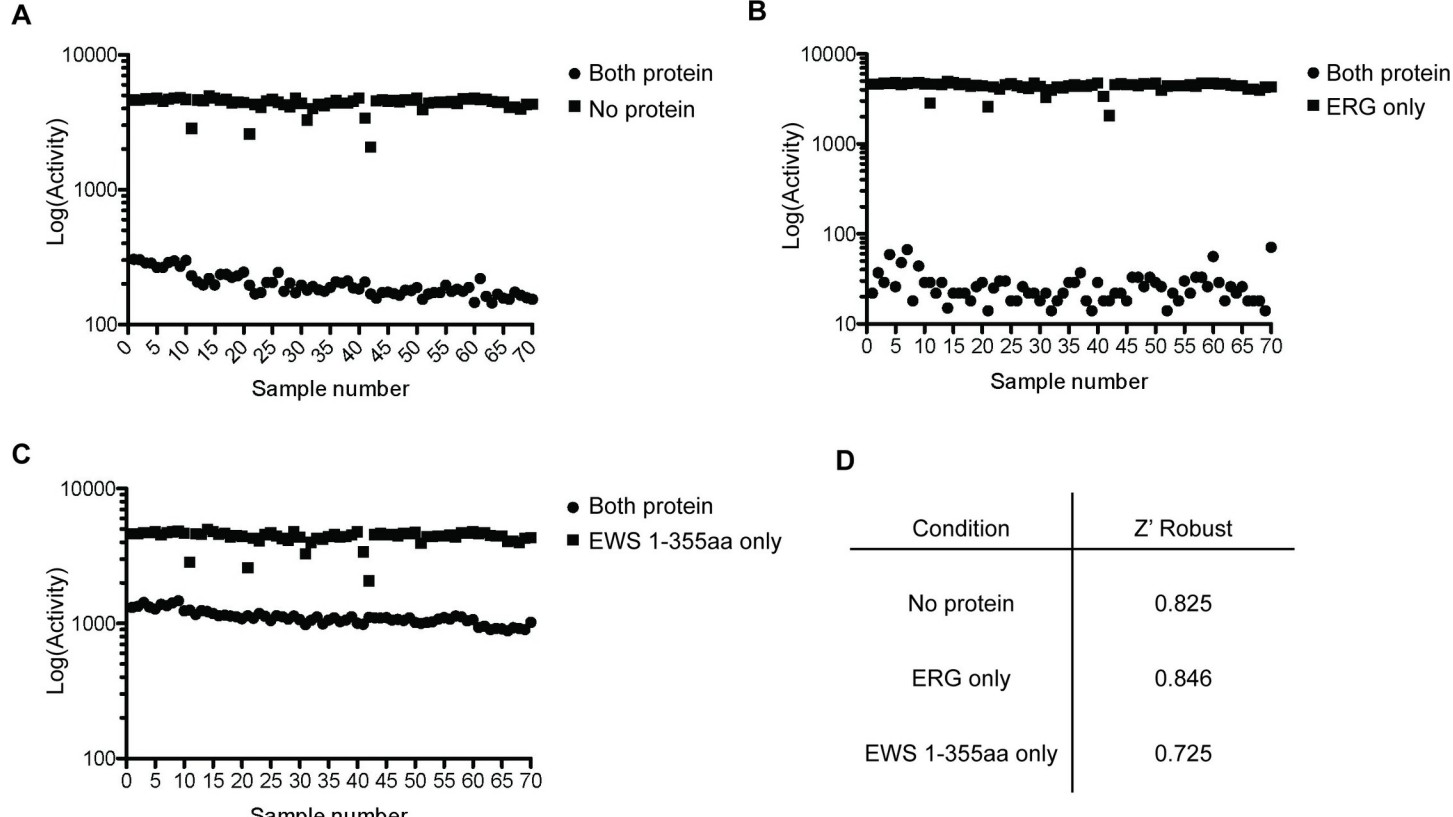

**Fig 3. Z' Robust determination.** A-C) Z' determination of the AlphaScreen system using beads alone without protein compared to both proteins (A), ERG conjugated to beads compared to both proteins with beads (B), and EWS 1-355aa conjugated to beads compared to both proteins with beads (C). D) Z' robust values for corresponding conditions.

purified ERG and EWS 1-355aa in assay buffer with DMSO and was used for background measurements for each plate. Column 22 contained protein-bead conjugates in assay buffer with DMSO and served as the positive control for each plate. The percent activity compared to the positive control was plotted, where percent activity is the signal minus background over the signal from the positive control multiplied by 100 (Fig 4A). Small molecules were called "hits" if they yielded a percent activity less than 35%. Eighteen hits were validated using the AlphaScreen in duplicate, along with two compounds that enhanced the protein-protein interaction (Fig 4B). From the 18 initial hits, six small molecules resulted in less than 10% percent activity compared to the DMSO control in the duplicate assay. Four of these six validated small molecules were commercially available, so these were chosen for hit-to-lead generation (Fig 4C).

## Hit-to-lead generation

The four hit small molecules were tested for the ability to inhibit the ERG-EWS protein-protein interaction in a secondary, fluorescent-independent assay. Importantly, we wanted to understand if the hits would inhibit the interaction between ERG and native full-length EWS. Purified His-ERG was bound to cobalt beads and used as bait to pull-down native EWS from prostate cancer cell line PC3 nuclear extract. Hits A, B, C, and D were added in the same concentration used in HTS, 62.5 µM. Compared to the DMSO control, treatment with Hit B or Hit D had the greatest effect (Fig 5A, upper). Several of the hit compounds are metal chelators that could interfere with the cobalt bead chemistry. However, ponceau staining indicated that none of the tested compounds decreased ERG binding to the beads (Fig 5A, lower). We then tested if lower concentrations of Hit B could disrupt the interaction. Inhibition of the ERG-EWS interaction by Hit B in two replicate experiments was dose-dependent, with slightly reduced inhibition at 45 µM and very little inhibition at 30 µM (Fig 5B).

We next wanted to test the ability of Hit B to inhibit known ERG-EWS-mediated cancer-associated phenotypes. We have previously shown that ectopic expression of ERG in an immortalized-normal prostate cell line (RWPE1) can drive cell migration, clonogenic survival, and target gene activation dependent on the interaction with EWS [9]. Importantly, the introduction of ERG does not alter RWPE1 viability or 2-D cell proliferation [9]. Interestingly, despite the high concentration, 45 µM Hit B had little effect on RWPE-ERG cell viability (Fig 5C). In contrast, 45 µM Hit B dramatically decreased the ERG-mediated phenotypes of cell migration (Fig 5D) and clonogenic growth (Fig 5E). In the case of cell migration, another compound, Hit D, had little effect at 45 µM, indicating that inhibition is a unique feature of Hit B.

Two assays were used to determine the half maximal inhibitory concentration (IC50) of Hit B on ERG function in RWPE1 cells. First cell migration (Fig 5F), and second a reporter assay (Fig 5G) where luciferase expression was driven by a known ERG *cis*-regulatory element, previously shown to require EWS co-activator function [9]. RWPE1-ERG cells were treated with a serial dilution of Hit B for each assay. The IC50 for Hit B was ~9 µM in both assays indicating that lower concentrations of the compound might be functional in cells than in the in vitro protein-protein interaction assay.

To test if the lead compounds alter protein stability or transactivation function, we used the VCaP prostate cancer cell line which harbors the *TMPRSS2/ERG* rearrangement. The *TMPRSS2/ERG* fusion in VCaP cells drives expression of an N-terminally truncated ERG protein. This truncated ERG binds the other, native ERG allele to drive expression of full length ERG [18]. Treatment of either Hit B or Hit D (45 µM) had no impact on EWS protein level as measured by immunoblot (Fig 5H, top panel). Treatment of Hit B, but not Hit D, reduced expression of full length ERG (Fig 5H, middle panel, top band) but not the truncated ERG

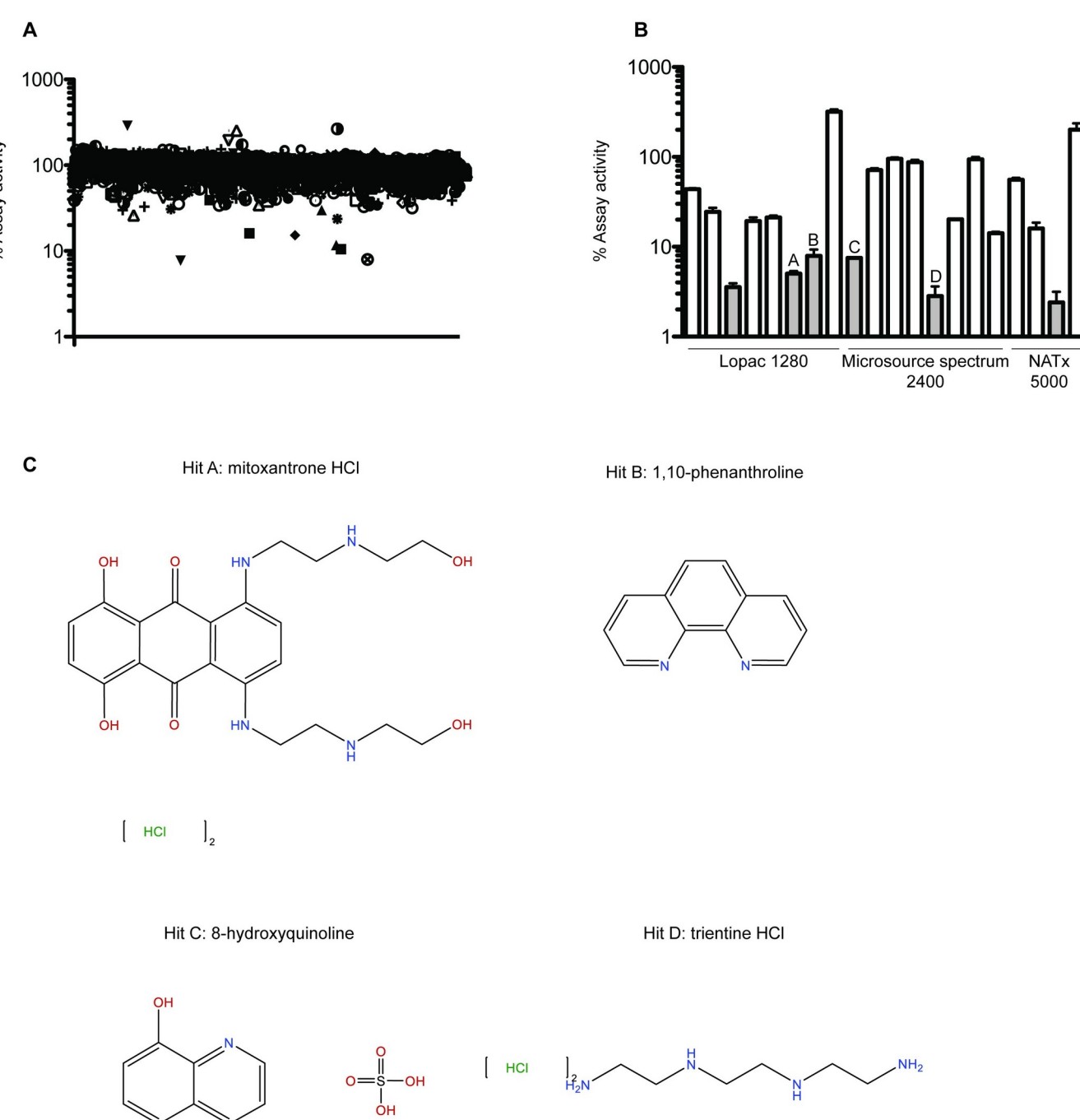

**Fig 4. AlphaScreen and validation.** A) Results of HTS. Percent assay activity compared to the DMSO control was plotted for the 8680 compounds tested. Compounds that resulted in less than 35% activity were considered initial hits. B) Validation of hits. Eighteen inhibitor hits and two protein-protein interaction enhancers were used in the AlphaScreen performed in duplicate and percent activity compared to the DMSO control was plotted. Libraries from which compounds are from are labeled below. C) Structures and hit ID of the four commercially available validated hits.

encoded by the TMPRSS2/ERG fusion (Fig 5H, middle panel, bottom band). These findings are consistent with Hit B altering the transcriptional activation function of ERG in VCaP cells. Therefore, we tested if Hit B could reduce mRNA levels of ERG target genes. Treatment of VCaP cells with 45 μM Hit B led to significant decreases in *ERG* and *HSPA8* mRNA

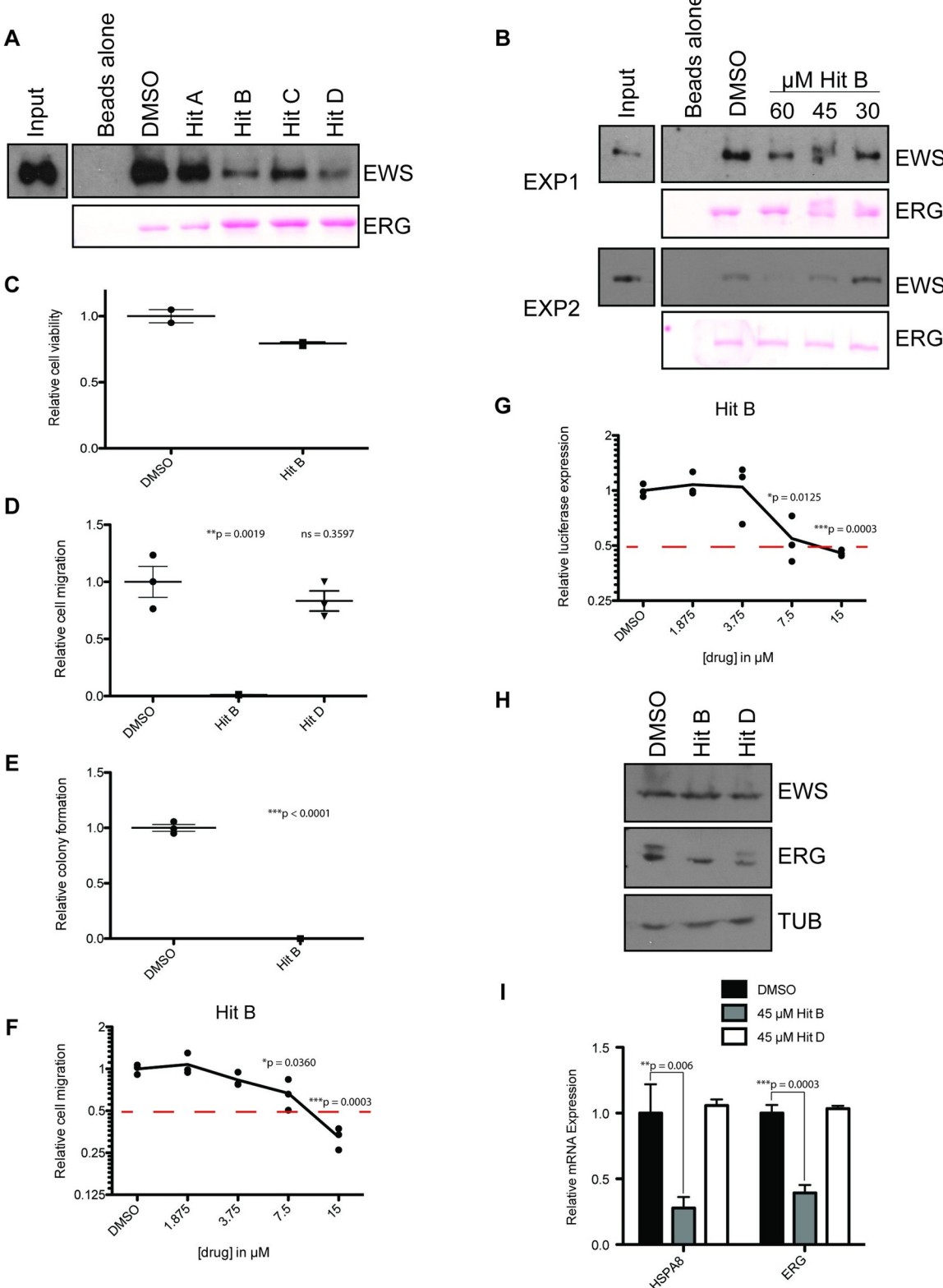

**Fig 5. Secondary assays determine lead compounds.** A) His-tag affinity pull-down validation of hits using PC3 nuclear extract which contains native EWS. Purified His-ERG was used as bait and ERG protein is shown by Ponceau stain (bottom). EWS signal is shown by EWS immunoblot (top). 62.5 μM of indicated Hit compounds were used. B) His-tag affinity pull-down as in (A) except using 60, 45, and

30 μM of Hit B. Two independent experiments are shown (EXP1, top; EXP2, bottom). C) Cell viability assay using RWPE1-ERG cells after treatment with 45 μM of indicated compound or DMSO. D) Cell migration assay using RWPE1-ERG cells after treatment with 45 μM of indicated compound or DMSO. E) Clonogenic growth assay using RWPE1-ERG cells after treatment with 45 μM of indicated compound or DMSO. F) IC50 determination by migration assay using RWPE1-ERG cells after treatment with DMSO or indicated compound concentrations in μM. The IC50, indicated by the point in which the red dashed line crosses the cell migration curve, is estimated to be ~9 μM. G) IC50 determination by luciferase assay. Luciferase expression is driven by an ETS-motif-containing enhancer element and firefly expression is normalized to renilla expression. IC50, indicated by the point in which the red dashed line crosses the relative luciferase curve, is estimated to be ~9 μM. H) Immunoblot of ERG and EWS from VCaP cells after treatment with 45 μM of indicated compound or DMSO for 24 hr. Tubulin (TUB) is used as a loading control. I) qRTPCR of known ERG target genes after treatment with 45 μM of indicated compound or DMSO for 24 hr in VCaP cells.

expression, measured by qRTPCR. In contrast, Hit D did not alter expression of *ERG* or *HSPA8* (Fig 5I).

## Discussion

In this study we find that AlphaScreen technology is a feasible HTS approach to identify small molecule inhibitors of ERG-EWS. We identified six validated hits from the AlphaScreen. Four of these compounds were available commercially, and two were able to decrease the ERG-EWS interaction using a secondary, non-fluorescent based protein-protein interaction assay. One hit compound, Hit B, was able to significantly inhibit the ERG-EWS mediated phenotypes cell migration, clonogenic growth, and reporter expression. Hit B was also able to decrease ERG target gene expression and expression of full length ERG protein. The concentrations necessary for inhibition were relatively high, however, these findings validate the usefulness of this screening technique to identify higher affinity compounds and indicate that Hit B could provide a lead compound for these efforts.

Hit B is 1,10-phenanthroline which has been previously studied as a possible cytotoxic therapeutic in liver and kidney cancer cell lines [19]. Our screen identified 1,10-phenanthroline as a compound that could disrupt the ERG/EWS protein-protein interaction that we have previously shown is necessary for ERG transactivation function. The immunoblot in Fig 5H supports a transcriptional inhibitory role, as 1,10 phenanthroline preferentially decreases full-length ERG compared to the truncated ERG encoded by *TMRPSS2/ERG*. ERG protein is involved in a positive feedback loop with the native ERG locus: both full length and truncated ERG bind the *ERG* promoter to active *ERG* transcription [18]. However, *TMPRSS2/ERG* is not ERG regulated since the native promoter is lost. To further support this result, *ERG* expression at the RNA level is significantly decreased by 1,10 phenantholine treatment, as well as another ERG downstream target, *HSPA8* (Fig 5I). However, these data do not rule out the possibility that 1,10 phenanthroline specifically alters full length ERG protein stability while not affecting truncated ERG. One limitation of this study is that engineered ERG was used to pull down EWS from cell extracts. Further testing is needed to verify that 1,10 phenanthroline can in fact inhibit the endogenous protein-protein interaction in cells, as well as *in vivo*.

IDRs often oscillate dynamically between structures. This is modulated by post-translational modifications and interactions with various macromolecules [2, 20]. Both the interacting interfaces of ERG and EWS are predicted to be intrinsically disordered, posing a challenge for the development of small molecule inhibitors based on structure alone. Therefore, many diverse small molecules must be systematically tested for the ability to interfere with this type of unstructured interface. It is possible to target IDRs with small molecules. Biotinylated isoxazole, or b-isox, is a small molecule that binds to IDRs in RNA binding proteins, including the N-terminus of EWS, that are associated with RNA granules [21]. B-isox binding disrupts granule formation by trapping the IDR in a structured state, suggesting that small molecule binding

to its target inhibits target function [21]. While B-isox lacks target specificity, it proves that IDRs can be targeted by small molecules.

This study is the first to screen for small molecule inhibitors of ERG-EWS. We screened 8680 molecules and found one candidate lead inhibitor, 1,10-phenanthroline. While 1,10-phenanthroline inhibited ERG-EWS at a high concentration, the AlphaScreen setup described here provides the basis and rationale for expansion of this screen using larger chemical libraries or derivative libraries synthesized from the lead compound identified in this study.

## Supporting information

**S1 File. Raw images.**
(PDF)

## Acknowledgments

The authors would like to acknowledge the Chemical Genomics Core Facility at Indiana University School of Medicine for expertise and use of equipment.

## Author Contributions

**Funding acquisition:** Peter C. Hollenhorst.

**Investigation:** Taylor R. Nicholas, Jingwei Meng, Benjamin M. Greulich, Teresa Stevie Morris.

**Supervision:** Peter C. Hollenhorst.

**Writing – original draft:** Taylor R. Nicholas.

**Writing – review & editing:** Peter C. Hollenhorst.

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
