## [Decision Letter · Decision Letter 0]

30 Jul 2020

PONE-D-20-18931

A high-throughput screen identifies inhibitors of the interaction between the oncogenic transcription factor ERG and the cofactor EWS

PLOS ONE

Dear Dr. Hollenhorst,

Thank you for submitting your manuscript to PLOS ONE. After careful consideration, we feel that it has merit but does not fully meet PLOS ONE’s publication criteria as it currently stands. Therefore, we invite you to submit a revised version of the manuscript that addresses the points raised during the review process.

Binding intrinsically disordered segments of transcription factors may lead to changes in either stability or transcriptional activity. Please determine whether treatment of RWPE1-ERG cells with hits B and D interfere with the stability or transcriptional activity of full length ERG using western blotting and expression of a couple of endogenous ERG target genes. Confirm this effect of the lead compound on ERG function in an independently derived cell line, such as VCaP.

Add description of origin and culture conditions for the RWPE1-ERG cell line to materials and methods section. Correct typos throughout the manuscript, such as “both protein” and some others. Conduct statistical analysis wherever differences are claimed and add statistical analysis description to the materials and methods section.

We look forward to receiving your revised manuscript.

Kind regards,

Irina U Agoulnik, Ph.D.

Academic Editor

PLOS ONE

Journal Requirements:

2.PLOS ONE now requires that authors provide the original uncropped and unadjusted images underlying all blot or gel results reported in a submission’s figures or Supporting Information files. This policy and the journal’s other requirements for blot/gel reporting and figure preparation are described in detail at https://journals.plos.org/plosone/s/figures#loc-blot-and-gel-reporting-requirements and https://journals.plos.org/plosone/s/figures#loc-preparing-figures-from-image-files. When you submit your revised manuscript, please ensure that your figures adhere fully to these guidelines and provide the original underlying images for all blot or gel data reported in your submission. See the following link for instructions on providing the original image data: https://journals.plos.org/plosone/s/figures#loc-original-images-for-blots-and-gels.

3. Please provide additional information about each of the RWPE1 cell line used in this work, including the culture conditions and any quality control testing procedures (authentication, characterisation, and mycoplasma testing). For more information, please see http://journals.plos.org/plosone/s/submission-guidelines#loc-cell-lines.

4. To comply with PLOS ONE submission guidelines, in your Methods section, please provide additional information regarding your statistical analyses. For more information on PLOS ONE's expectations for statistical reporting, please see https://journals.plos.org/plosone/s/submission-guidelines.#loc-statistical-reporting.

Reviewers' comments:

Reviewer's Responses to Questions

**Comments to the Author**

1. Is the manuscript technically sound, and do the data support the conclusions?

Reviewer #1: Yes

2. Has the statistical analysis been performed appropriately and rigorously? 

Reviewer #1: I Don't Know

3. Have the authors made all data underlying the findings in their manuscript fully available?

Reviewer #1: Yes

4. Is the manuscript presented in an intelligible fashion and written in standard English?

Reviewer #1: Yes

5. Review Comments to the Author

Reviewer #1: This manuscript by Nicholas et al. reports small molecules identified to inhibit the oncogenic ERG-EWS interaction. Authors previously identified ERG-EWS interaction as a driving factor of tumor growth in prostate cancer. Authors utilized AlphaScreen for highthrouput analysis and tested prostate cancer phenotypes in cultured cells to validate the molecules. One candidate lead inhibitor was identified from this scan. The manuscript is straight forward written and appropriate methods with adequate controls are used.

There are minor issues that need to be addressed before it is ready for publication. This is not the first study aimed at targeting an ERG protein-protein interaction for the development of a small molecule-based prostate cancer therapy (PMID: 27223260). Authors may have to tone down such claims. Authors aim to demonstrate "Interestingly, an N-terminal degradation product arising from full-length ERG, and about the same size as EWS 1-355, also interacted with ERG better than full-length EWS. This suggests that the interaction with the N-terminus may be autoinhibited by the C-terminal region." however, the presented data does not fully support this conclusion. Authors acknowledge that wild type EWS interacting with rearranged ERG as an oncogenic mechanism. Would ERG fusion difference in cell lines used affect ERG/EWS interaction. What is the rational behind using PC3 nuclear extract for CoIP? How is AR dependence of cell lines affect ERG-EWS interaction? Authors need to acknowledge the limitations of the study such as engineered protein used in the assays. The reviewer was not able to judge the statistic rigor of the analyses.

6. PLOS authors have the option to publish the peer review history of their article (what does this mean?). If published, this will include your full peer review and any attached files.

Reviewer #1: **Yes: **Paul Basil

---

## [Author Response · Author response to Decision Letter 0]

26 Aug 2020

Binding intrinsically disordered segments of transcription factors may lead to changes in either stability or transcriptional activity. Please determine whether treatment of RWPE1-ERG cells with hits B and D interfere with the stability or transcriptional activity of full length ERG using western blotting and expression of a couple of endogenous ERG target genes. Confirm this effect of the lead compound on ERG function in an independently derived cell line, such as VCaP.

The authors would like to very much thank the editor for these helpful suggestions, as they led to a result that nicely supports our hypothesis. As suggested, we used VCaP cells. In 2011, Mani et al. showed that the TMPRSS2/ERG fusion in VCaP cells encodes an N-terminally truncated version of ERG that can bind to the promoter of the other allele of full-length ERG and activate it. We treated VCaP cells with DMSO, hit B, or hit D and measured protein levels of ERG and EWS by immunoblot. We found that expression of EWS was unchanged while levels of full-length, but not TMRPSS2/ERG encoded ERG protein were decreased by Hit B, but not Hit D. This would be consistent with what we would expect if Hit B can disrupt the transcriptional activation activity of ERG.

We then looked at expression of two known ERG target genes in VCaP cells after treatment. Both HSPA8 and ERG expression at the RNA level were significantly decreased upon treatment with hit B. Treatment with hit D had no impact. These new data are in the revised Figure 5.

Add description of origin and culture conditions for the RWPE1-ERG cell line to materials and methods section. Correct typos throughout the manuscript, such as “both protein” and some others. Conduct statistical analysis wherever differences are claimed and add statistical analysis description to the materials and methods section.

A description of origin and culture conditions for the RWPE1-ERG cell line was added to the materials and methods section under “Cell Culture and Antibodies”. Typos have been corrected. Statistical analyses were added when appropriate and a “Statistical Analysis” paragraph was added to the materials and methods section.

Our manuscript now meets the PLOS ONE style requirements.

2.PLOS ONE now requires that authors provide the original uncropped and unadjusted images underlying all blot or gel results reported in a submission’s figures or Supporting Information files. This policy and the journal’s other requirements for blot/gel reporting and figure preparation are described in detail at https://journals.plos.org/plosone/s/figures#loc-blot-and-gel-reporting-requirements and https://journals.plos.org/plosone/s/figures#loc-preparing-figures-from-image-files. When you submit your revised manuscript, please ensure that your figures adhere fully to these guidelines and provide the original underlying images for all blot or gel data reported in your submission. See the following link for instructions on providing the original image data: https://journals.plos.org/plosone/s/figures#loc-original-images-for-blots-and-gels.

 We have provided the original uncropped and unadjusted images in the supplemental figure file S1_File.pdf

3. Please provide additional information about each of the RWPE1 cell line used in this work, including the culture conditions and any quality control testing procedures (authentication, characterisation, and mycoplasma testing). For more information, please see http://journals.plos.org/plosone/s/submission-guidelines#loc-cell-lines.

 Cell culture conditions have been updated in the Materials and methods section. We have also detailed how cell lines have been authenticated and tested for mycoplasma.

4. To comply with PLOS ONE submission guidelines, in your Methods section, please provide additional information regarding your statistical analyses. For more information on PLOS ONE's expectations for statistical reporting, please see https://journals.plos.org/plosone/s/submission-guidelines.#loc-statistical-reporting.

The authors have added a “Statistical analysis” section to the material and methods.

Reviewers' comments:

Reviewer's Responses to Questions

Comments to the Author

5. Review Comments to the Author

Reviewer #1: This manuscript by Nicholas et al. reports small molecules identified to inhibit the oncogenic ERG-EWS interaction. Authors previously identified ERG-EWS interaction as a driving factor of tumor growth in prostate cancer. Authors utilized AlphaScreen for highthrouput analysis and tested prostate cancer phenotypes in cultured cells to validate the molecules. One candidate lead inhibitor was identified from this scan. The manuscript is straight forward written and appropriate methods with adequate controls are used.

There are minor issues that need to be addressed before it is ready for publication. This is not the first study aimed at targeting an ERG protein-protein interaction for the development of a small molecule-based prostate cancer therapy (PMID: 27223260). Authors may have to tone down such claims. Authors aim to demonstrate "Interestingly, an N-terminal degradation product arising from full-length ERG, and about the same size as EWS 1-355, also interacted with ERG better than full-length EWS. This suggests that the interaction with the N-terminus may be autoinhibited by the C-terminal region." however, the presented data does not fully support this conclusion. Authors acknowledge that wild type EWS interacting with rearranged ERG as an oncogenic mechanism. Would ERG fusion difference in cell lines used affect ERG/EWS interaction. What is the rational behind using PC3 nuclear extract for CoIP? How is AR dependence of cell lines affect ERG-EWS interaction? Authors need to acknowledge the limitations of the study such as engineered protein used in the assays. The reviewer was not able to judge the statistic rigor of the analyses.

The authors would like to thank this reviewer for a thoughtful review. We will address each comment with our response below:

This is not the first study aimed at targeting an ERG protein-protein interaction for the development of a small molecule-based prostate cancer therapy (PMID: 27223260). Authors may have to tone down such claims.

Thanks to the reviewer for pointing out that the BRD4 inhibitors JQ1 and iBET762 are reported to inhibit the protein-protein interaction between BRD4 and ERG. We have toned down our claim.

Authors aim to demonstrate "Interestingly, an N-terminal degradation product arising from full-length ERG, and about the same size as EWS 1-355, also interacted with ERG better than full-length EWS. This suggests that the interaction with the N-terminus may be autoinhibited by the C-terminal region." however, the presented data does not fully support this conclusion.

We agree that the data inferred such conclusion but did not directly support the claim. We therefore toned down our claim.

Authors acknowledge that wild type EWS interacting with rearranged ERG as an oncogenic mechanism. Would ERG fusion difference in cell lines used affect ERG/EWS interaction.

In our previous publication (PMID: 27783944) we found that the first 275 amino acids of ERG are not necessary for the interaction with EWS, and identified a region in the C-terminus of ERG that is necessary for the interaction. All known ERG fusion gene products have this region and would be predicted to bind EWS equally. 

What is the rational behind using PC3 nuclear extract for CoIP?

PC3 nuclear extract is purchased from Santa Cruz Biotechnology and is an economical source of native EWS protein. 

How is AR dependence of cell lines affect ERG-EWS interaction?

AR dependence of cell lines does not appear to impact the ERG-EWS interaction. The protein-protein interaction is maintained in VCaP cells (androgen sensitive) and RWPE1-ERG cells (androgen insensitive). The interaction is also maintained in primary patient tumors. These data are provided in our previous publication (PMID: 27783944).

Authors need to acknowledge the limitations of the study such as engineered protein used in the assays.

We have now acknowledged this limitation in the discussion.

The reviewer was not able to judge the statistic rigor of the analyses.

The authors have added statistical analyses to appropriate figures and added a “Statistical analysis” section to the material and methods.

The authors would like to thank this reviewer for a thoughtful review. We will address each comment with our response below:

This is not the first study aimed at targeting an ERG protein-protein interaction for the development of a small molecule-based prostate cancer therapy (PMID: 27223260). Authors may have to tone down such claims.

Thanks to the reviewer for pointing out that the BRD4 inhibitors JQ1 and iBET762 are reported to inhibit the protein-protein interaction between BRD4 and ERG. We have toned down our claim.

Authors aim to demonstrate "Interestingly, an N-terminal degradation product arising from full-length ERG, and about the same size as EWS 1-355, also interacted with ERG better than full-length EWS. This suggests that the interaction with the N-terminus may be autoinhibited by the C-terminal region." however, the presented data does not fully support this conclusion.

We agree that the data inferred such conclusion but did not directly support the claim. We therefore toned down our claim.

Authors acknowledge that wild type EWS interacting with rearranged ERG as an oncogenic mechanism. Would ERG fusion difference in cell lines used affect ERG/EWS interaction.

In our previous publication (PMID: 27783944) we found that the first 275 amino acids of ERG are not necessary for the interaction with EWS, and identified a region in the C-terminus of ERG that is necessary for the interaction. All known ERG fusion gene products have this region and would be predicted to bind EWS equally. 

What is the rational behind using PC3 nuclear extract for CoIP?

PC3 nuclear extract is purchased from Santa Cruz Biotechnology and is an economical source of native EWS protein. 

How is AR dependence of cell lines affect ERG-EWS interaction?

AR dependence of cell lines does not appear to impact the ERG-EWS interaction. The protein-protein interaction is maintained in VCaP cells (androgen sensitive) and RWPE1-ERG cells (androgen insensitive). The interaction is also maintained in primary patient tumors. These data are provided in our previous publication (PMID: 27783944).

Authors need to acknowledge the limitations of the study such as engineered protein used in the assays.

We have now acknowledged this limitation in the discussion.

The reviewer was not able to judge the statistic rigor of the analyses.

The authors have added statistical analyses to appropriate figures and added a “Statistical analysis” section to the material and methods.

---

## [Editor Report · Decision Letter 1]

28 Aug 2020

A high-throughput screen identifies inhibitors of the interaction between the oncogenic transcription factor ERG and the cofactor EWS

PONE-D-20-18931R1

Dear Dr. Hollenhorst,

We’re pleased to inform you that your manuscript has been judged scientifically suitable for publication and will be formally accepted for publication once it meets all outstanding technical requirements.

Kind regards,

Irina U Agoulnik, Ph.D.

Section Editor

PLOS ONE
---

## [Editor Report · Acceptance letter]

3 Sep 2020

PONE-D-20-18931R1 

­­A high-throughput screen identifies inhibitors of the interaction between the oncogenic transcription factor ERG and the cofactor EWS 

Dear Dr. Hollenhorst:

I'm pleased to inform you that your manuscript has been deemed suitable for publication in PLOS ONE. Congratulations! Your manuscript is now with our production department. 

Kind regards, 

on behalf of

Dr. Irina U Agoulnik 

Academic Editor

PLOS ONE